# The Use of Selected Ion Flow Tube-Mass Spectrometry Technology to Identify Breath Volatile Organic Compounds for the Detection of Head and Neck Squamous Cell Carcinoma: A Pilot Study

**DOI:** 10.3390/medicina55060306

**Published:** 2019-06-25

**Authors:** Dhinashini Chandran, Eng H. Ooi, David I Watson, Feruza Kholmurodova, Simone Jaenisch, Roger Yazbeck

**Affiliations:** 1Discipline of Surgery, College of Medicine and Public Health, Flinders University, Adelaide 5042, Australia; eooi.entsurgery@gmail.com (E.H.O.); david.watson@flinders.edu.au (D.I.W.); simone.jaenisch@flinders.edu.au (S.J.); roger.yazbek@flinders.edu.au (R.Y.); 2Flinders Centre for Innovation in Cancer, Flinders University, Adelaide 5042, South Australia; 3Flinders Center for Epidemiology and Biostatistics, Flinders University, Adelaide 5042, South Australia; feruza.kholmurodova@flinders.edu.au

**Keywords:** breath test, head and neck cancer, neoplasms head and neck, cancer screening, cancer screening tests, volatile organic compounds

## Abstract

*Background:* Head and neck squamous cell carcinoma (HNSCC) is the sixth most common form of cancer worldwide, with approximately 630,000 new cases diagnosed each year. The development of low-cost and non-invasive tools for the detection of HNSCC using volatile organic compounds (VOCs) in the breath could potentially improve patient care. The aim of this study was to investigate the feasibility of selected ion flow tube mass spectrometry (SIFT-MS) technology to identify breath VOCs for the detection of HNSCC. *Materials and Methods:* Breath samples were obtained from HNSCC patients (N = 23) and healthy volunteers (N = 21). Exhaled alveolar breath samples were collected into FlexFoil^®^ PLUS (SKC Limited, Dorset, UK) sampling bags from newly diagnosed, histologically confirmed, untreated patients with HNSCC and from non-cancer participants. Breath samples were analyzed by Selected Ion Flow Tube-Mass Spectrometry (SIFT-MS) (Syft Technologies, Christchurch, New Zealand) using Selective Ion Mode (SIM) scans that probed for 91 specific VOCs that had been previously reported as breath biomarkers of HNSCC and other malignancies. *Results:* Of the 91 compounds analyzed, the median concentration of hydrogen cyanide (HCN) was significantly higher in the HNSCC group (2.5 ppb, 1.6–4.4) compared to the non-cancer group (1.1 ppb, 0.9–1.3; Benjamini–Hochberg adjusted p < 0.05). A receiver operating curve (ROC) analysis showed an area under the curve (AUC) of 0.801 (95% CI, 0.65952–0.94296), suggesting moderate accuracy of HCN in distinguishing HNSCC from non-cancer individuals. There were no statistically significant differences in the concentrations of the other compounds of interest that were analyzed. *Conclusions:* This pilot study demonstrated the feasibility of SIFT-MS technology to identify VOCs for the detection of HNSCC.

## 1. Introduction

Head and neck squamous cell carcinoma (HNSCC) accounts for more than 630,000 cases and 330,000 deaths worldwide each year [1]. There is currently no useful validated method for detecting HNSCC. Generally the detection and diagnosis of HNSCC relies on specialist surgical assessment and panendoscopy with tissue biopsy performed under general anesthesia [2].

Breath analysis represents a non-invasive, rapid, and painless technique that has been investigated for the early detection of cancers [3,4] and infection [5]. A breath-based screening tool could potentially be used by primary care providers for initial investigation when seeing at risk patients for HNSCC. Breath analysis could be used to identify and target patients requiring urgent imaging and expedite referral pathways to a specialist oncology surgeon. A breath analysis tool could also help in the surveillance of patients after treatment of their head and neck cancer. The concept of non-invasive monitoring is demonstrated by the urea carbon-14 breath test to detect the presence of *Helicobacter pylori* and to check that the bacteria has been eradicated in patients after treatment.

Human breath contains a complex mixture of gases that includes oxygen, nitrogen, hydrogen, and trace amounts of volatile organic compounds (VOCs) [6]. Identified as the chemical byproducts of normal cell metabolism, VOCs are present in exhaled breath and bio fluids such as blood, saliva, sweat, and urine [7]. Emerging research has identified breath VOCs as potential non-invasive biomarkers for HNSCC [3,4,8,9,10,11,12], breast [13,14,15,16], gastro-esophageal [17], and lung cancers [18,19,20].

Recent studies, using technologies such as Electronic Nose (E-Nose) [9,12], Gas Chromatography-Mass Spectrometry (GC-MS) [3,4], Proton Transfer Reaction-Mass Spectrometry (PTR)-MS [10], and Solid-Phase Microextraction (SPME)-MS [8] in HNSCC have demonstrated the diagnostic utility of breath VOCs to distinguish patients with HNSCC from non-cancer individuals. The aim of this pilot study was to investigate the use of Selected Ion Flow Tube-Mass Spectrometry (SIFT-MS) technology to identify breath volatile organic compounds for the detection of head and neck squamous cell carcinoma.

## 2. Materials and Methods

This was a prospective cohort study comparing exhaled breath compounds in patients with HNSCC to exhaled breath compounds from non-cancer individuals. Ethics approval was obtained from the Southern Adelaide Local Health Network Human Ethics Committee (HREC/16/SAC/70 66.16) on the 29th of April 2016. (Site Specific Application (SSA) reference number is SSA/16/SAC/76.). Participants were recruited through the Department of Otolaryngology and Head and Neck Surgery at Flinders Medical Centre (FMC) and the Royal Adelaide Hospital (RAH), from May 2016 to November 2016.

### 2.1. Patient Recruitment

The HNSCC group was defined as patients with histologically confirmed SCC of the larynx, hypopharynx, oropharynx, and oral cavity prior to beginning treatment for their cancer. Participants in the non-cancer group were over 18 years of age and consisted of healthy volunteers with no previous history of cancer.

Participants with recurrent HNSCC, post-treatment of prior HNSCC, or concurrent other cancers were excluded from this study. Those with diabetes mellitus were also excluded, as breath biomarkers of oxidative stress have been found to be significantly elevated in the breath of individuals with type 1 and 2 diabetes mellitus [21]. All participants were asked to complete a questionnaire which included; age, gender, weight, medical history, smoking and alcohol history, and dietary history. Smoking status was defined as never smoked, ex-smoker, or current smoker.

### 2.2. Breath Collection and Sampling

Breath samples were collected into 4 ply 3 L FlexFoil (SKC Ltd., Pennsylvania, USA) breath analysis bags via a single polypropylene fitting. Before use, bags were flushed with nitrogen gas a minimum of seven times to remove phenol and other background VOCs associated with the bags. A new FlexFoil bag was used for each participant. The FlexFoil sample bags were chosen due to the sample storage stability for up to 24 h and low background VOC content. A room air sample was collected concurrently with each breath sample to account for non-endogenous VOCs that may be present in the environment. Room air samples were collected using a gas tight glass syringe, and then transferred to a flushed Flex Foil bag using polytetrafluoroethylene (PTFE) tubing.

All participants were required to fast and abstain from smoking for a minimum of 6 h prior to breath collection. Breath collection was performed in seated position via a single deep nasal inhalation, followed by complete exhalation through their mouth into the collection bag. FlexFoil bags were transported in a thermal insulator bag for immediate transport and analysis in the breath laboratory. In situations where more than one breath sample was collected from multiple participants on the day, the sealed bags were stored in a 37 °C dry incubator for a maximum of 3 h before analysis by SIFT-MS.

### 2.3. Breath Analysis

Breath samples were analyzed by SIFT-MS (Syft Technologies, Christchurch, New Zealand) using a full spectral MASS scan. This mass scan was set to measure all 723 product ions with masses ranging from 10 to 250 atomic mass units (amu) over 50 cycles. Selective Ion Mode (SIM) scans were then used to probe for 91 specific VOCs that had been previously reported as breath biomarkers of HNSCC and other malignancies (Table 1). The product ion masses of interest were first selected from the Syft Compound Library. SIM scans were further refined by eliminating compounds with low reaction rates and abundances (branching ratios) and those that conflicted with other target compounds. In order to maximize the sensitivity of the target compound quantitation, we aimed to retain the maximum number of non-conflicting product masses within a SIM scan. The concentrations of the VOCs of interest (in parts per billion volume) were then statistically analyzed to identify compounds that were significantly different between the HNSCC and non-cancer groups.

### 2.4. Statistical Analysis

Statistical analysis was performed with the IBM SPSS statistics version 20 (SPSS Inc, Chicago, IL, USA) and Stata version 15.1 (Stata Corp, College Station, Texas, USA). Data was expressed as medians and interquartile ranges, or frequencies, as appropriate. Demographics, smoking, and alcohol-related data between the groups were analyzed using the Mann–Whitney U and Chi square tests. The SIM scan data was analyzed with the Mann–Whitney U test. Type I error from multiple hypothesis testing was addressed with False Discovery Rate correction using the Benjamini–Hochberg adjusted p-value cutoff of ≤ 0.05. The performance of discriminating VOCs was assessed using the area under the receiver operating characteristic (ROC) curve. Logistic regression analysis was used to evaluate the association between HNSCC and HCN and was adjusted for age and gender. Results are reported as odds ratios and 95% confidence intervals (CI). The type 1 error rate was set at p < 0.05.

## 3. Results

### 3.1. Demographics and Patient Characteristics

Breath samples were collected from patients with HNSCC (N = 23) and non-cancer volunteers (N = 21). The baseline characteristics of the participants are summarized in Table 2. There were significant differences in the distribution of age, gender, smoking, and alcohol consumption between the two groups. The HNSCC group was predominantly male and significantly older than the non-HNSCC group (Table 2). The median smoking pack years and alcohol intake per week were significantly higher amongst the patients with HNSCC compared to the non-cancer subjects (Table 2). The HNSCC sub-sites used in this study were the oral cavity (N = 4), oropharynx (N = 8), larynx (N = 8), and hypopharynx (N = 3). The stages (AJCC 7th edition) according to tumor site are summarized in Table 3. Oropharyngeal and laryngeal SCCs were the most common within the cohort.

### 3.2. Breath VOCs Identified by SIM Scan

The 91 volatile organic compounds (VOCs) that had been previously associated with HNSCC and other malignancies were quantified in the breath of HNSCC and non-cancer groups by SIM scan methods. The median concentration of HCN was significantly higher in the HNSCC group (4.62 ppb, 2.21–7.82) compared to the non-cancer group (3.73 ppb, 1.46–6.80; Benjamini–Hochberg adjusted p ≤ 0.05 at FDR of 6%) (Appendix A). When HCN was measured specifically in room air samples, the median concentration was 2.92 ppb (1.51–3.95). No other differences in breath VOC concentrations were found between groups.

### 3.3. The Accuracy and Effectiveness of HCN as a Biomarker for HNSCC

A basic receiver operating curve (ROC) analysis was performed to investigate the optimum combination of sensitivity and specificity of HCN as a potential biomarker of HNSCC. This showed an area under the curve (AUC) of 0.801 (95% CI, 0.65952–0.94296), suggesting moderate accuracy of HCN in distinguishing HNSCC from non-cancer individuals (Figure 1). The cut-off value, negative predictive value (NPV), positive predictive value (PPV), sensitivity, and specificity are presented in Table 4. The area under the curve (AUC) for adjusted analysis was 0.849. Odds ratios were borderline significant for HCN and gender (female as reference), OR = 1.85 (p = 0.050) and OR = 5.68 (p = 0.041) respectively.

## 4. Discussion

This pilot study is the first study to demonstrate that SIFT-MS technology can identify VOCs in HNSCC and non-HNSCC patients. SIFT-MS is an analytical technique that was developed by David Smith and Patrik Spanel for precise and accurate real time quantification of trace gases in air [25]. SIFT-MS technology involves chemical ionization of the breath sample with precursor ions, H_3_O^+^, NO^+^, or O_2_^+^ [25]. This reaction results in the formation of product ions, which are analyzed by a mass spectrometer to identify and quantify VOCs [25]. The ability to provide instant quantification of all the analytes in a sample is a major beneficial advantage of SIFT-MS. Furthermore, since its inception, SIFT-MS has evolved from a large lab-based instrument to a smaller, readily transportable instrument [26]. SIFT-MS has promising potential to be used in the clinical setting, owing to its rapid ability to identify and quantify specific VOCs.

To date, various other breath analysis technologies have been used in the detection of HNSCC. These techniques include off-line sampling analysis via Gas Chromatography-Mass Spectrometry (GC-MS) [4], and other real-time analytical techniques such as PTR-MS [10] and E-Nose [9]. The E-Nose, used in the majority of HNSCC studies published to date [9,11,12], is based on a combination of chemical sensor arrays and pattern recognition algorithms and thus, cannot distinguish specific VOCs. Unfortunately, the target VOCs used to build the E-Nose, are not reported in the literature, preventing direct comparison with our study. GC-MS, the second most common technology used in similar studies, has high sensitivity and compound resolution [27].

Our results show that SIFT-MS technology could be used to identify potential breath biomarkers for the non-invasive detection of HNSCC. Using SIFT-MS SIM scan analysis, HCN was found at significantly higher concentrations in the breath samples of HNSCC patients compared to non-cancer controls. There were no other differences found between groups for the other breath VOCs measured in this study. VOCs previously associated with HNSCC in the literature [3,4,8] (mainly hydrocarbons, alkanes, alkanes, and nitriles) were not elevated above the levels of the non-cancer group in this study, which could be attributed to several factors, including different study methodology, patient groups, analytical technologies, breath sampling techniques, patient demographics, and geographical factors. Our findings using SIFT-MS add to the growing body of evidence on the detection of VOCs in HNSCC. Over the last three decades, several studies have also demonstrated the diagnostic potential of breath VOCs in the detection of lung, breast, colorectal, esophageal, and gastric cancers [23]. A recently published systematic review and meta-analysis on the accuracy of phase 1 biomarker studies that identified potential VOCs in the detection of various cancers (breast, thyroid, lung, gastric, head and neck, mesothelioma, colorectal, and esophagogastric) reported an accuracy of 0.94 (AUC), sensitivity of 79% (95% CI, 77–81%), and specificity of 89% (95% CI, 88–90%) [24]. This study suggested standardizing breath collection and validation methods in future multicenter trials.

The finding of elevated HCN in the HNSCC compared to the non-cancer group is a novel finding. HCN is found both in the environment and in human breath. In the environment, it is mainly a product of biomass burning, industrial processes, vehicle exhaust, and cigarette smoke [28]. HCN is also produced through several biological processes in plants, fruits, and vegetables [28]. In humans, HCN is present at low (parts per billion) volumes in the exhaled breath of the general population and has been associated with metabolic pathways affecting cellular respiration in the human body [29]. Cyanides are well absorbed via the gastrointestinal tract or skin and rapidly metabolized via the respiratory tract. Approximately 80% of absorbed HCN is metabolized to thiocyanate in the liver by mitochondrial sulfur transferase enzymes [28]. Mutations in mitochondrial DNA affecting cytochrome oxidase of the cellular respiratory complex have been reported in HNSCC patients; however, the exact role of mitochondrial mutations in the development of HNSCC is unknown [30]. In our study, exogenous compounds from the diet and cigarette smoke were minimized by fasting and abstaining from cigarettes and alcohol for a minimum of 6 h prior to breath collection. The 3 L FlexFoil bags allowed for a mixed alveolar air sample to be collected in a single, forced exhalation, minimizing the concentration of the dead space air component. Room air samples were also collected from all locations where breath sampling was performed to account for ambient air contaminants. We speculate that the higher levels of HCN in HNSCC found in this study could be related to this enzymatic pathway through mitochondrial DNA mutations. Hence, the link between HCN and cytochrome oxidase in the potential pathogenesis of HNSCC is an area that could be explored further.

HCN in exhaled breath may have also been produced by changes in the oral microbiome. In humans, HCN may be produced by microorganisms such as proteobacteria and fungi and is thought to be derived from the metabolism of methylene carbon of glycine [31]. Elevated HCN levels have been detected in the breath of adults and children with cystic fibrosis and *Pseudomonas aeruginosa* infection [31]. In HNSCC, high rates (32.7%) of *Pseudomonas bacteremia* have been reported in patients with oral cavity SCCs undergoing chemo-radiotherapy [32]. Hence, it could be speculated that an altered microbiome with *Pseudomonas aeruginosa* presence in HNSCC patients could, in part, explain the observation of elevated HCN in HNSCC patients.

Age, gender, and smoking habits were significantly different between the HNSCC and non-cancer groups in this study. This is because HNSCC is primarily a disease which involves an older male population with major risk factors of tobacco and alcohol consumption [33]. Matching of participants to age, gender, and smoking proved to be challenging, as the majority of the HNSCC patients were above the age of 55 years old, predominantly male, and heavy smokers compared to the non-cancer group. The majority of HNSCC patients in our pilot study lived alone and belonged to a lower socioeconomic group. Hence, matching of dietary and lifestyle factors by using the spouse as a non-cancer control was difficult to achieve. Furthermore, there is insufficient evidence to suggest that the patient characteristics of age and gender could have affected the breath VOC profiles in this study [29,34]. Future clinical studies with a larger number of participants and incorporating regression models will be needed to accurately investigate the impact of patient characteristics, smoking, stage, tumor sub-site, and HPV on breath VOC profiles in HNSCC.

This pilot study has demonstrated the feasibility of breath testing in a clinical environment using SIFT-MS. However, the small sample size and patient heterogeneity, particularly in smoking habits, tumor subsite, and stages of HNSCC, are acknowledged as limitations of this study. The challenge associated with matching smoking habits between non-cancer and HNSCC patient groups has been discussed above. Heterogeneity in tumor subsite and stage were accepted as potential confounding factors. The association between HCN and patient characteristics were not evaluated due to the small sample size. Hence, we are cautious to propose that HCN alone is suitable as a detection breath biomarker for HNSCC due to the small sample size and limitations in terms of matching the patient group to a non-cancer cohort. However, our study shows that SIFT-MS technology can be used to identify breath VOCs in HNSCC patients.

## 5. Conclusions

Breath biomarkers are a promising non-invasive approach for the detection of HNSCC and surveillance after treatment. Further research with larger patient numbers and non-cancer subjects more closely matched to the patients is required to gain a deeper understanding of the confounding effects of patient characteristics to aid the development of a population-based detection test for HNSCC.

## Figures and Tables

**Figure 1 medicina-55-00306-f001:**
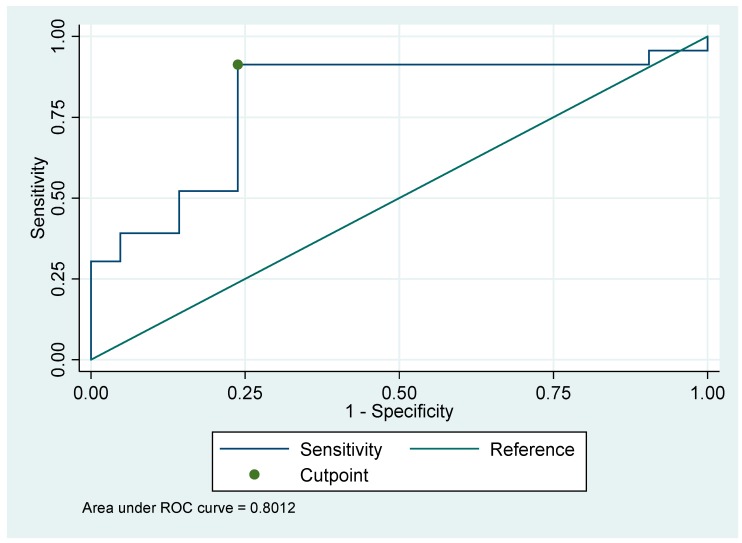
Receiver operating curve (ROC) curve for the ability of HCN to discriminate between the breath of Head and Neck Squamous Cell Carcinoma (HNSCC) and non-cancer individuals.

**Table 1 medicina-55-00306-t001:** VOCs selected for SIM scan analysis of breath samples.

**HNSCC associated VOCs** 2,2-dimethylpropanoic acid ethanol undecane limonene 1-octene butanone acrylonitrile **Colorectal Cancer associated VOCs** isopentane (colorectal cancer) 2-methoxy-2-methyl butane 2-methylbutanoic acid decanal (colorectal cancer) N2 isopentane methylcyclohexane methylcyclopentane **Breast cancer associated VOCs** 2-propanol (breast cancer) isobutanoic acid (breast cancer) acetic acid (breast cancer) N2 2-methylundecanal tridecane longifolene cyclopropane 1,3-butadiene 1,4-benzoquinone **Gastric cancer associated VOCs** methyl n-propyl sulfide pentanal (gastric cancer) pentanoic acid furfural 4-methyloctanoic acid nonanal octanal	**Lung Cancer associated VOCs** 3-methylhexane methanol p-xylene methanol hexan-2-ol 1-propanol acetophenone benzoic acid cyclohexanone dodecane benzene 2-methylpentane 3-hexanone propanoic acid decane toluene 1,2,4-trimethylbenzene methylcyclopentane propylbenzene heptanal pentane hexanal acetone heptanal styrene decane isoprene formaldehyde isooctane furan N2 m-xylene 2,3-butanediol camphor ethane 3-methylbutanoic acid 3-methylhexane acetophenone methyl isobutyl ketone	**Liver Disease associated VOCs** dimethyl sulfide formic acid methyl mercaptan **Infection associated VOCs** ammonia hydrogen cyanide indole **Halitosis** isopropylamine trimethylamine 3-methylindole 1,5-diaminopentane Freon 113 acetoin 1,6-dihydrocarveol N2 1,1-dichloroethane **Inflammatory bowel disease associated VOCs** hydrogen sulfide dimethyl disulfide ammonia N2 butanoic acid Isobutanoic acid 1,4-diaminobutane

The SIM scans consisted of VOCs that had been previously linked with cancers and other biological processes such as infection and inflammatory processes [22,23,24]. Head and Neck Squamous Cell Carcinoma (HNSCC), Volatile Organic Compounds (VOCs), Selective Ion Mode (SIM).

**Table 2 medicina-55-00306-t002:** Demographics and patient characteristics.

Factor	Level	Non-Cancer	HNSCC	p-Value
N		21	23	
P16 status (positive)		-	9	
Age, median (IQR)		52 (41,60)	61.5 (52.5,65.3)	0.030 ^†^
Gender	F	13 (62%)	3 (13%)	<0.001 ^*^
	M	8 (38%)	20 (87%)	
Height (cm), median (IQR)		165.5 (162.4,172.0)	172.0 (162.0,180.0)	0.25
Weight (kg), median (IQR)		75.0 (68.3,84.0)	73.0 (61.0,93.0)	0.91
Smoking Status	Never	9 (43%)	3 (13%)	0.057
	Ex	8 (38%)	10 (43%)	
	Current	4 (19%)	10 (43%)	
Smoking (pack years), median (IQR)		1.0 (0.0,20.0)	39.0 (25.0,52.0)	<0.001 ^§^
Alcohol intake (AUS standard drink)/week, median (IQR)		1.0 (0.0,4.0)	6.0 (0.0,21.0)	0.019 ^§^
Hydrogen cyanide (74-90-8), median (IQR)		1.1 (0.9,1.3)	2.5 (1.6,4.4)	<0.001 ^§^

Values expressed as n (%), except where otherwise indicated. ^a^ Values expressed as median (range). HNSCC = head and neck squamous cell carcinoma; U = unit of alcohol, 1 unit is approximately 10 g of alcohol. ^†^ t test, ^§^ Mann–Whitney U Test, ^*^ Chi square test.

**Table 3 medicina-55-00306-t003:** HNSCC subsites and stages.

Cancer Site	Number of Cases	Stage	
		1/11	111/1V
Oral Cavity	4	3	1
Oropharynx	8	6	2
Larynx	8	4	4
Hypopharynx	3	1	2

**Table 4 medicina-55-00306-t004:** Diagnostic discrimination analysis of HCN.

Cut-off Value	AUC	Sensitivity (%)	Specificity (%)	NPV (%)	PPV (%)
1.25	0.801	91	76	88.6	80.4

AUC = Area under the curve; NPV = Negative Predictive Value; PPV = Positive Predictive Value.

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
