# Peer review of "The Use of Selected Ion Flow Tube-Mass Spectrometry Technology to Identify Breath Volatile Organic Compounds for the Detection of Head and Neck Squamous Cell Carcinoma: A Pilot Study"

_medicina, 2019, doi:10.3390/medicina55060306_

Round 1
Reviewer 1 Report
The authors have addressed one of my concerns by including the data for HCN in the main body of the paper.
However though they say in their response to reviewers that they have included data for all other VOCs in supplementary tables, I can't see any mention of this in the revised manuscript.
Once this is addressed then I would be happy to recommend for publication
Author Response
We have attached two supplementary tables
-Supplementary Table 1. VOC data for Healthy Group
-Supplementary Table 2. VOC data for Cancer Group

Reviewer 2 Report
The manuscript by Chandran et al. uses Ion Flow Tube-Mass Spectrometry technology to analyze 91 VOCS in 23 head and neck squamous cell carcinoma patients and 21 controls.
The 91 VOCs analyzed in the present study and listed in Table 1 are generically indicated as previously reported cancer- and HNSCC–related VOCs and some of them seem to be linked to other biological process. This assertion needs to be detailed and quoted. The HNSCC – and cancer - related VOCs need to be clearly indicated in Table 1.
The consistence of the VOCs profiles from subjects sampled in two different hospitals should be evaluated to detect and correct the possible presence of batch effect in the data. How many patients and controls were sampled in each hospital?
Author Response
Comment 1: The manuscript by Chandran et al. uses Ion Flow Tube-Mass Spectrometry technology to analyze 91 VOCS in 23 head and neck squamous cell carcinoma patients and 21 controls. The 91 VOCs analyzed in the present study and listed in Table 1 are generically indicated as previously reported cancer- and HNSCC–related VOCs and some of them seem to be linked to other biological process. This assertion needs to be detailed and quoted. The HNSCC – and cancer - related VOCs need to be clearly indicated in Table 1.
Response: We have now detailed and quoted cancer and HNSCC related VOCs in Table 1. This has been revised in the attached manuscript.
Comment 2: The consistence of the VOCs profiles from subjects sampled in two different hospitals should be evaluated to detect and correct the possible presence of batch effect in the data. How many patients and controls were sampled in each hospital?
Response: Only 6 breath samples from cancer patients were collected from patients located at the Royal Adelaide Hospital (RAH), whilst 17 breath samples were collected from patients attending the Flinders Medical Centre (FMC). All control samples were collected from Flinders University.
The median HCN levels in the FMC and RAH groups were 3.81 [range: 2.21, 4.39] and 1.62 [range: 0.85,1.63]) respectively. While the median HCN in the RAH group is lower, there is quite a large variation between patients. Furthermore, only n=6 was collected from the RAH, so it's unlikely to have a significant impact or batch effect.

Reviewer 3 Report
The author has addressed my concern by logical explanation.
Author Response
Response:
Thank you.